# MicroRNA-9 downregulates the ANO1 chloride channel and contributes to cystic fibrosis lung pathology

Florence Sonneville[1], Manon Ruffin[1], Christelle Coraux[2], Nathalie Rousselet[1], Philippe Le Rouzic[1], Sabine Blouquit-Laye[3], Harriet Corvol[1,4] & Olivier Tabary[1]

Cystic fibrosis results from reduced cystic fibrosis transmembrane conductance regulator protein activity leading to defective epithelial ion transport. $Ca^{2+}$-activated $Cl^-$ channels mediate physiological functions independently of cystic fibrosis transmembrane conductance regulator. Anoctamin 1 (ANO1/TMEM16A) was identified as the major $Ca^{2+}$-activated $Cl^-$ channel in airway epithelial cells, and we recently demonstrated that downregulation of the anoctamin 1 channel in cystic fibrosis patients contributes to disease severity via an unknown mechanism. Here we show that microRNA-9 (miR-9) contributes to cystic fibrosis and downregulates anoctamin 1 by directly targeting its 3′UTR. We present a potential therapy based on blockage of miR-9 binding to the 3′UTR by using a microRNA target site blocker to increase anoctamin 1 activity and thus compensate for the cystic fibrosis transmembrane conductance regulator deficiency. The target site blocker is tested in in vitro and in mouse models of cystic fibrosis, and could be considered as an alternative strategy to treat cystic fibrosis.

[1] Centre de Recherche Saint Antoine (CRSA), INSERM, Sorbonne Universités, UPMC Univ Paris 06, F75012 Paris, France. [2] INSERM UMR-S 903, University of Reims Champagne-Ardenne, 51100 Reims, France. [3] Université de Versailles Saint Quentin en Yvelines, UFR des Sciences de la Santé, 78180 Montigny-Le-Bretonneux, France. [4] Paediatric Respiratory Department, Hôpital Trousseau, AP-HP, 75012 Paris, France. Correspondence and requests for materials should be addressed to O.T. (email: olivier.tabary@inserm.fr)

Cystic fibrosis (CF) affects multiple organs, but morbidity and death in CF patients are caused mainly by chronic bacterial lung infection and inflammation leading to progressive lung damage[1]. CF is caused by mutations in the CF transmembrane conductance regulator (*CFTR*) gene that encodes a chloride channel. The CFTR protein localizes to the apical membrane of epithelial cells and participates in fluid and electrolyte homeostasis[2].

While current therapies are mainly symptomatic, various drugs targeting the CFTR protein itself have been recently developed[3]. However, functional CFTR rescue remains limited, and the effects are mutation-dependent[4, 5]. Therefore, to compensate for the CFTR deficiency, alternative strategies have been suggested, such as the stimulation of calcium-activated chloride channels (CaCCs); however, the identity of these chloride channels has remained elusive for over 20 years[6]. In 2008, three research teams independently identified anoctamin 1 (ANO1) or transmembrane protein 16a (TMEM16A) as a CaCC, which they proposed as a potential CF therapeutic target[7–9]. Interestingly, ANO1 shows the same expression pattern as CFTR at the apical membrane of epithelial cells. Moreover, ANO1 is involved in HCO3− permeability, proliferation, wound healing, inflammation, and cell migration, which are deregulated in CF[10–13]. We recently showed that ANO1 expression, migration, and activity are decreased in CF patients[14]. In the current study, we explored the mechanisms involved in the downregulation of ANO1 with the aim to enhance its expression and function as an alternative therapeutic strategy for CF.

MicroRNAs (miRNAs) are evolutionarily conserved non-coding RNAs that negatively regulate gene expression by repressing translation or decreasing mRNA stability. Only a few groups have studied the role of miRNAs in CF, and they mainly explored the relationship between CFTR and miRNA expression[15]. Therefore, we investigated the role of miRNAs, especially, miR-9, on ANO1 expression in bronchial epithelial cell lines and primary highly differentiated cells cultured in an air–liquid interface.

We show that miR-9 is overexpressed in CF cells where it directly regulates ANO1, causing a decrease in its expression and activity. Based on this knowledge, we propose an alternative strategy to correct chloride efflux in CF patients that rely on a target site blocker (TSB) that prevents miR-9 targeting of ANO1. This therapeutic approach results in increased chloride efflux mediated by ANO1, mucociliary clearance, and migration rate of cells in in vitro and in vivo CF models.

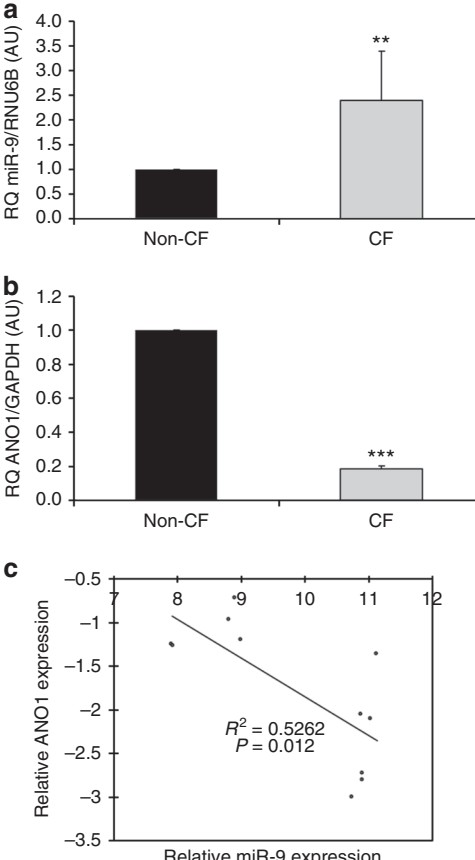

**Fig. 1** Correlation of downregulation of ANO1 mRNA and upregulation of miR-9. **a** miR-9 expression in non-CF (16HBE14o-; $n = 5$) and CF (CFBE41o-; $n = 6$) bronchial epithelial cell lines measured by qRT-PCR. Relative expression levels were normalized to those of RNU6B. Data are presented as the mean ± SD and were compared using Student's *t*-test. All qRT-PCR experiments were performed in triplicate. **b** Relative expression levels of ANO1 mRNA in non-CF (16HBE14o-; $n = 5$) and CF (CFBE41o-; $n = 6$) human bronchial epithelial cells normalized to GAPDH. Data are quantified by qRT-PCR and are presented as a fold-change compared to normalized controls. Data are presented as the mean ± SD and were compared using Student's *t*-test. All qRT-PCR experiments were performed in triplicate. **c** Pearson's correlation analysis showed a negative correlation between miR-9 and ANO1 mRNA expression levels in non-CF and CF bronchial epithelial cell lines ($P = 0.012$)

## Results

**Upregulation of miR-9 correlates with downregulation of ANO1.** We previously showed that ANO1 expression and activity are decreased in CF, but the mechanisms involved remained unknown[14]. To predict ANO1-targeting miRNAs involved in these decreases, we used several computational tools, including Targetscan, miRDB, miRANDA, and Pictar. Four miRNAs were predicted to target ANO1: miR-9, miR-19a, miR-19b, and miR-144. After preliminary luciferase assays to test the binding of these miRNAs to the ANO1 3′UTR, we focused on miR-9 (see Supplementary Figs. 1–6 for results on the other miRNAs). miR-9 expression was studied in CF (CFBE41o-) and non-CF (16HBE14o-) bronchial epithelial cells by qRT-PCR and showed a 2.5-fold increase in CF cells as compared to non-CF cells (Fig. 1a). In contrast, ANO1 expression was decreased by 80% in CF cells (Fig. 1b). Interestingly, ANO1 and miR-9 transcript levels were significantly inversely correlated in our cell line models ($P = 0.012$; Pearson's correlation) (Fig. 1c). This result was confirmed in primary human bronchial glandular cells ($P <$

0.003; Pearson's correlation) and in fully differentiated bronchial cells from CF patients cultured in an air–liquid interface (ALI) ($P < 0.05$; Pearson's correlation; see Supplementary Fig. 7a, b).

**miR-9 regulates ANO1 expression and chloride activity.** The increase in miR-9 expression in CF cells led us to hypothesize that ANO1 is a direct target of miR-9. We transfected non-CF cells (16HBE14o-) with a miR-9 mimic and verified the transfection efficiency by qRT-PCR and by using a SmartFlare probe (Supplementary Figs. 8 and 9). Additionally, we confirmed the specificity of the miR-9 mimic by quantifying the expression of other miRNAs (including miR-19a) by qRT-PCR (Supplementary Fig. 10). Then, we assessed the effects of miR-9 overexpression on ANO1 mRNA and protein expression. Non-CF cells were transfected with a miR-9 mimic or a negative control. miR-9 overexpression significantly decreased ANO1 mRNA expression by 60% on average (Fig. 2a). Western blotting indicated that the

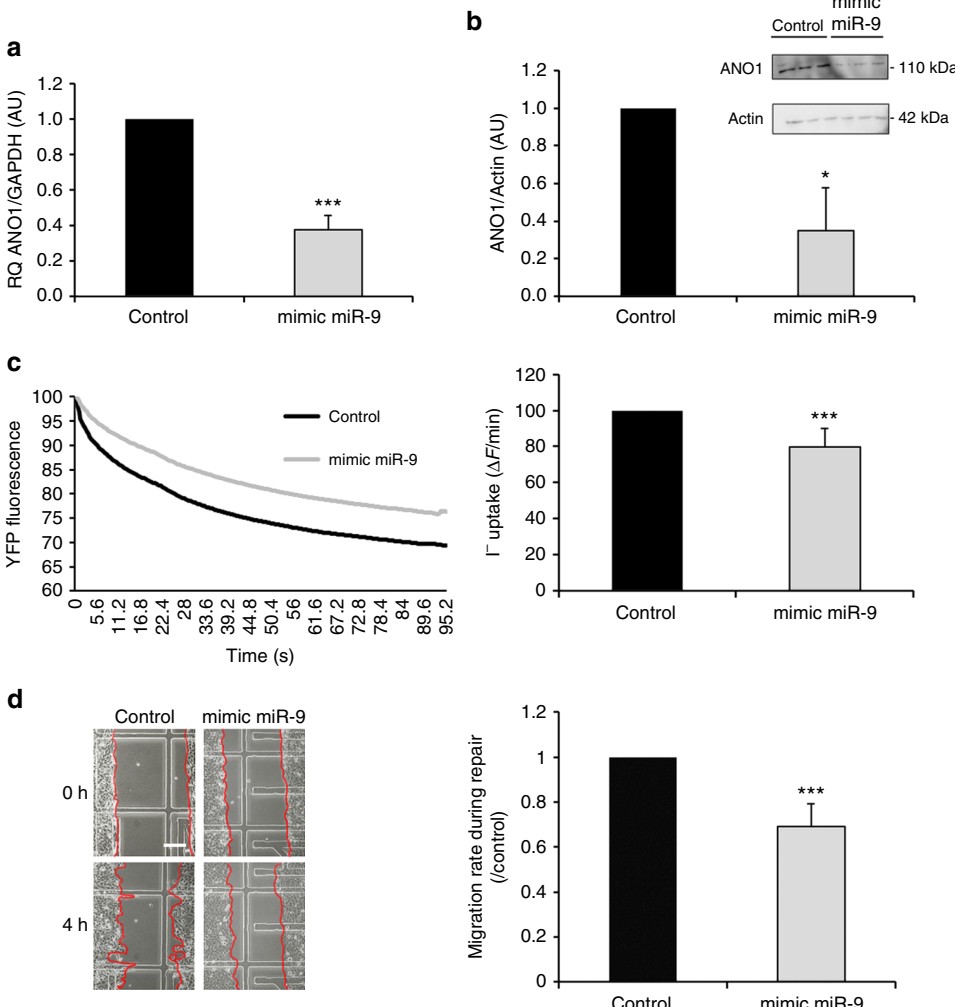

**Fig. 2** miR-9 regulates ANO1 expression, ANO1 chloride activity, and non-CF cells migration. Non-CF cells (16HBE14o-) were transfected with a miR-9 mimic (30 nM) or a negative control for 48 h. **a** ANO1 mRNA expression was analyzed by RT-qPCR and normalized to GAPDH ($n = 3$). Data are presented as the mean $\pm$ SD and were compared using Student's $t$-test. All qRT-PCR experiments were performed in triplicate. **b** ANO1 protein expression was analyzed by western blotting using anti-ANO1 antibody and normalized to β-actin ($n = 3$, in triplicates). Data are presented as the mean $\pm$ SD and were compared using Student's $t$-test. **c** ANO1 chloride channel activity assessed by I⁻ quenching of halide-sensitive YFP-H148q/I152L protein. Representative and original traces of ANO1 chloride activity (*left*) and quantification (*right*) of non-CF cells transfected with a miR-9 mimic or negative control ($n = 8$, in triplicates). Data are presented as the mean $\pm$ SD and were compared using Student's $t$-test. **d** Representative images were taken during 4 h of wound closure of non-CF cells transfected with a miR-9 mimic or a negative control (*left*) and quantification of the migration rates during repair ($n = 5$). Data are presented as the mean $\pm$ SD and were compared using Student's $t$-test

ANO1 protein level significantly decreased after transfection of the cells with a miR-9 mimic (Fig. 2b). Thus, we concluded that miR-9 negatively regulates ANO1 expression in our model of non-CF human bronchial epithelial cells.

Next, we evaluated the effect of miR-9 on ANO1 channel activity, and we have shown that miR-9 overexpression led to a significant decrease in ANO1 channel activity (Fig. 2c). Because we had previously demonstrated that ANO1 is involved in CF and non-CF cell migration[14], we studied the effect of miR-9 overexpression on 16HBE14o- migration. To avoid cell proliferation and to focus on cell migration alone, we have conducted the experiment for 4 h only as from that time point onward; we detected a significant increase in proliferation (Supplementary Fig. 11). Transfection of 16HBE14o- with a miR-9 mimic led to a 30% decrease in the migration rate compared to cells transfected with a negative control- (Fig. 2d). These results suggest that miR-9 regulates ANO1 chloride activity and cell migration in non-CF human bronchial epithelial cells.

**miR-9 directly regulates ANO1 in bronchial epithelial cells**. To test whether miR-9 represses ANO1 expression by binding to the ANO1 3′UTR, 16HBE14o- cells were cotransfected with a miR-9 mimic and a luciferase reporter vector containing WT ANO1 3′ UTR (WT-ANO1 3′UTR) or a negative control reporter harboring mutations in the predicted miR-9-binding sites (3′UTR mut ANO1). Cotransfection with miR-9 mimic resulted in a significant 40% reduction of luciferase gene expression from WT-ANO1 3′UTR as compared to 3′UTR mut ANO1 demonstrating direct miR-9-ANO1 interaction in non-CF cells (Fig. 3a). When the same experiment was conducted in CFBE41o- CF cells using an inhibitor of miR-9, we observed a significant increase in luciferase activity in cells transfected with the inhibitor as compared to those transfected with the control, whereas no significant difference was observed with cells harboring the mutated plasmid (Fig. 3b). CF cells transfected with a miR-9 mimic exhibit a significant decrease of luciferase-3′UTR ANO1 activity and non-CF cells transfected with a miR-9 inhibitor exhibit a significant

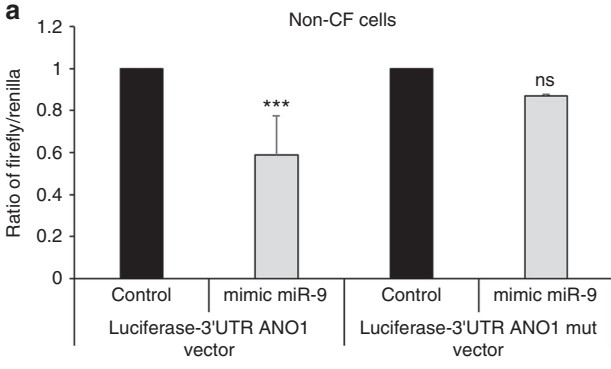

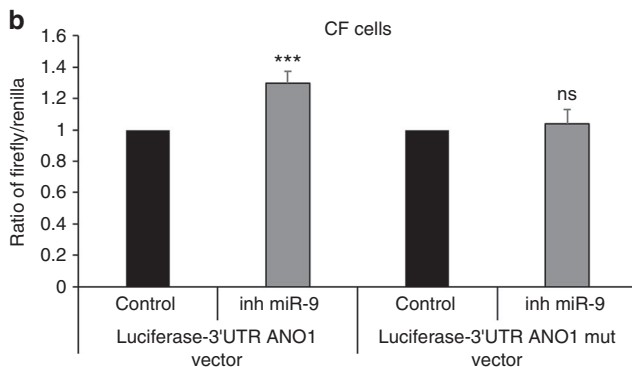

**Fig. 3** miR-9 directly targets ANO1 3′UTR in non-CF and CF cells. **a** Relative luciferase activity in non-CF cells (16HBE14o-) transiently transfected with a luciferase-3′UTR ANO1 vector or a luciferase-3′UTR ANO1 vector mutated at miR-9-binding sites and co-transfected with a miR-9 mimic or a negative control (control). Firefly luciferase activity was normalized to *Renilla* luciferase activity ($n = 3$, with 8 replicates). Histograms represent average values ± SDs and were compared using Student's *t*-test. **b** Relative luciferase activity in CF cells (CFBE41o-) transiently transfected with luciferase-3′UTR ANO1 or luciferase-3′UTR ANO1 mutated at miR-9-binding sites and cotransfected with an inhibitor of miR-9 (inh miR-9) or a negative control (control). Firefly luciferase activity was normalized to *Renilla* luciferase activity ($n = 3$, with 8 replicates). Histograms represent average values ± SDs and were compared using Student's *t*-test

increase (Supplementary Fig. 12a, b). We concluded that miR-9 directly regulates ANO1 in CF cells.

**miR-9 specific TSB increases ANO1 expression and chloride activity.** In the context of CF, the therapeutic goal is to enhance ANO1 chloride channel activity. As miR-9 has multiple targets, miR-9 inhibitors cannot be utilized to prevent specifically ANO1 downregulation. Thus, we designed a specific TSB that binds to the ANO1 3′UTR (ANO1 TSB) to prevent miR-9 binding. The TSB combines different technologies (locked nucleic acid [LNA] and phosphorothioates), allowing enhanced stability, cell permeability, and specificity[16]. We transfected CF cells with either miRCURY LNA negative control (TSB control) or ANO1 TSB for 24 h and then quantified ANO1 protein. ANO1 protein expression was increased 2.5-fold ($P < 0.05$; Student's *t*-test) in the cells transfected with ANO1 TSB (Fig. 4a). By using microinjection of a fluorescent reporter in association with microscopy, we were able to observe a single cell microinjected with the TSB control and another cell microinjected with ANO1 TSB within the same field (Supplementary Movies 1 and 2; Supplementary Figs. 13 and 14). The results revealed a larger chloride efflux in the ANO1 TSB-transfected cells indicating increased ANO1 channel activity (Fig. 4b). In a more classical approach, we quantified ANO1 channel activity using the Premo Halide Sensor method, which

confirmed the increase in ANO1 channel activity in the cells transfected with ANO1 TSB ($P < 0.05$; Student's *t*-test) (Fig. 4c). Interestingly, in cells transfected with ANO1 TSB, the ANO1 channel activity was similar to that observed in non-CF cells. The specificity of ANO1 activity was confirmed in an experiment using specific siRNA-ANO1[14] and by sh-ANO1 in a CF cell line (Supplementary Fig. 15). Finally, the cell migration rate in cells transfected with ANO1 TSB was significantly higher than that in TSB control cells (Fig. 4d). Together, these results indicated that ANO1 TSB could specifically target ANO1 and modulate its channel activity and that it plays a role in cell migration.

**TSB increases deficient parameters in primary human CF cells.** To mimic in vivo epithelium, we used primary human bronchial epithelial cells (hAECB) isolated from bronchial biopsies from CF (F508del/F508del) patients and fully differentiated ALI cultures of these cells. We successfully transfected the cells by adding medium containing ANO1 TSB or TSB control to the apical face of ALI cultures, without the use of any transfection reagent (Fig. 5a, b; Supplementary Movie 3). After 2 h incubation at 37 °C, the medium was removed from the apical face to restore the ALI conditions. Freshly prepared control oligonucleotides or ANO1 TSB were added on three consecutive days, and the effects of the transfection were observed 24 h post treatment. Western blot results showed a significant 68% increase in ANO1 expression in ANO1 TSB-transfected cells as compared to the control (Fig. 5c), while the chloride channel activity was increased 1.7-fold in these cells (Fig. 5d). Further, we noted a 2.2-fold increase in the migration rate of primary CF cells transfected with ANO1 TSB (Fig. 5e; Supplementary Movies 4 and 5). Additionally, we studied the mucus dynamics using the movement of fluorescent beads (Fig. 5f; Supplementary Movies 6 and 7). Interestingly, the average speed of movement of the beads on ANO1 TSB-expressing cells was higher than that on TSB control-treated cells. Thus, the utilization of ANO1 TSB in primary CF cells allowed improving the chloride activity, migration rate, and mucus dynamics of the cells, suggesting that ANO1 TSB has therapeutic potential for patients.

**TSB increases chloride activity and mucus dynamic in CF mice.** As the miR-9 sequence and the miR-9 seed sequence in the 3′ UTR of ANO1 are very well conserved (Supplementary Figs. 16 and 17), we performed experiments in immortalized mouse lung epithelial cells (MLE-15). For TSB control and ANO1 TSB transfections, we used the same protocol as for the human cell lines, and we quantified ANO1 expression (Supplementary Fig. 18) and channel activity (Fig. 6a) as previously described[14]. In vitro, MLE-15 cells treated with ANO1 TSB exhibited a strong (3-fold) increase in channel activity (Fig. 6a). Because of the promising results in vitro, we next evaluated the effects of ANO1 TSB in CF mice. To determine the potency and efficacy of ANO1 TSB in vivo, male CF (CFTR[tm1Eur]) mice received control or ANO1 TSB via intranasal instillations at a previously reported effective dose[17]. The mice were treated once every week and were killed on day 21 (Fig. 6b). Weight gains were recorded to estimate any toxic effects of the intranasal instillation and/or TSB. No weight losses were observed after intranasal instillation of ANO1 TSB or control, suggesting that the TSB is well tolerated by the mice (Fig. 6c). Upon killing of the mice, the tracheas were collected for inflammatory and chloride channel activity analyses. We focused on the trachea because ANO1 expression is important in this tissue[18] and because mice trachea closely resembles human bronchial tissue. No inflammatory cytokines (IL-1β, KC, or IL-6) were differentially expressed in the trachea or lung of the mice suggesting limited toxicity of the TSB (Supplementary

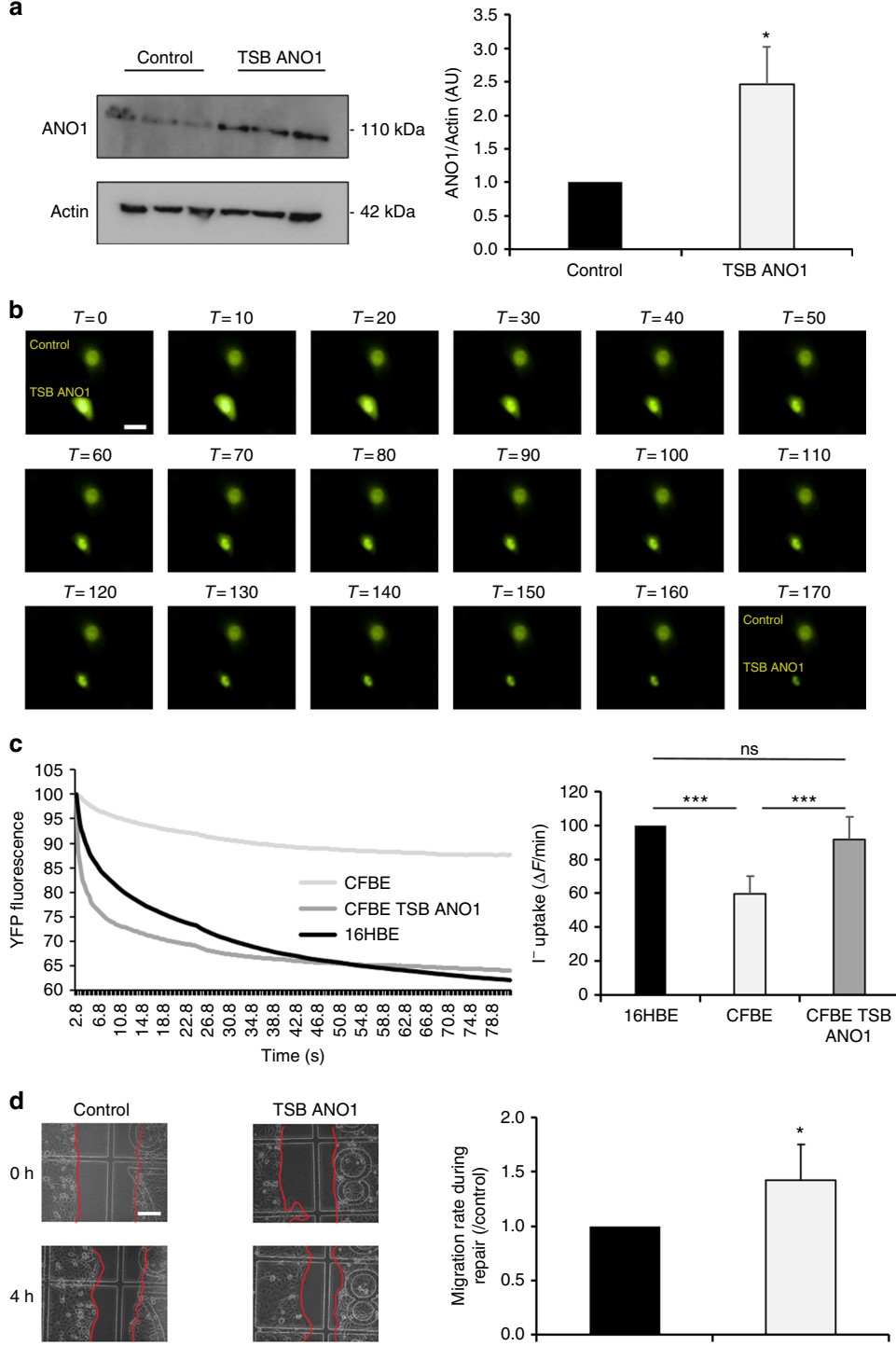

**Fig. 4** miR-9-specific TSB increases ANO1 expression, chloride activity, and migration rate. CF cells (CFBE41o-) were transfected with TSB control (control) or ANO1 TSB for 24 h. **a** ANO1 protein expression was analyzed and quantified by western blotting using anti-ANO1 antibody. β-actin was used for normalization. ($n = 4$, in triplicates). Histograms represent average values ± SDs and were compared using Student's $t$-test. **b** Kinetics of YFP-H148Q/I152L protein quenching after addition of I$^-$ to the medium. Twenty-four hours after transfection with YFP-H148Q/I152L plasmid, cells were selectively microinjected with TSB control or ANO1 TSB as indicated. *Scale bar* 5 μm. **c** Representative and original traces of ANO1 channel activity (*left*) and quantification (*right*) of CF bronchial epithelial cells transfected with ANO1 TSB or a negative control ($n = 8$, in triplicates) as compared to non-CF cells (16HBE14o-). Histograms represent average values ± SDs and the conditions were compared using ANOVA coupled with Dunnett's, Bonferroni's and Tukey's post hoc test. **d** Migration rates during repair of CF cells. Representative images were taken during 4 h of wound closure of CF cells (*left*) and migration rates were quantification (*right*) ($n = 5$). *Scale bar* 10 μm. Histograms represent average values ± SDs and were compared using Student's $t$-test

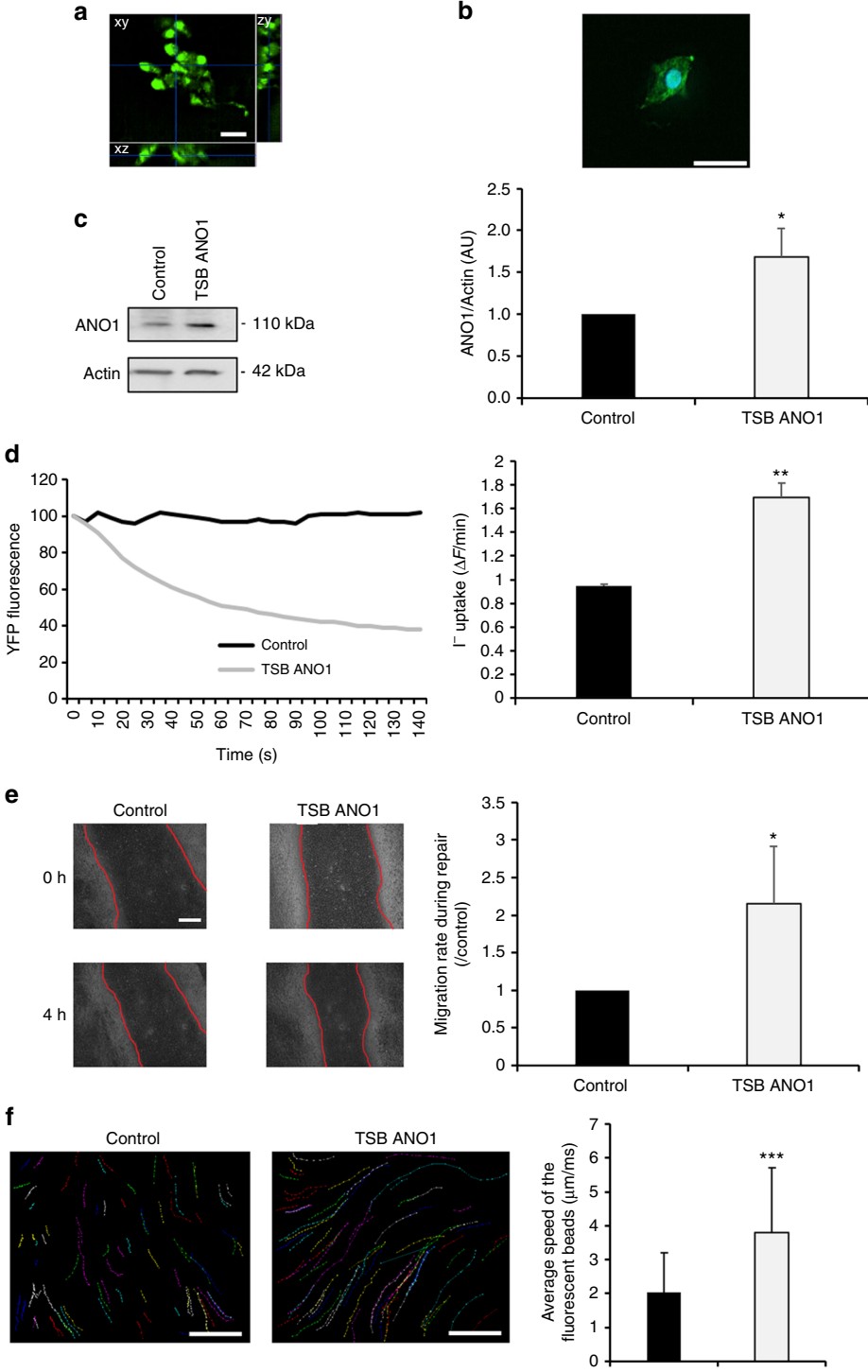

**Fig. 5** miR-9-specific TSB restores deficient parameters in primary CF cells. Primary hAECB and fully differentiated human bronchial air–liquid-interface cultures, isolated from bronchial biopsies from CF (F508del/F508del) patients were transfected with TSB control (control) or ANO1 TSB every day during 3 days. **a** Confocal microscopic analysis of fluorescein-conjugated TSB transfected into human bronchial cells isolated from CF patients (*green*). Cells were cultured in ALI and transfected for 24 h. **b** Isolated cells from ALI cultures transfected with fluorescein-conjugated TSB (*green*). The nuclei were stained with DAPI (*blue*), and merged images are shown. *Scale bars* 10 μm. **c** ANO1 protein expression was analyzed and quantified by western blotting using anti-ANO1 antibody. β-actin was used for normalization. Histograms represent average values ± SDs ($n = 6$) and were compared using Student's $t$-test. **d** Representative and original traces of ANO1 channel activity (*left*) and quantification (*right*) of hAECB CF cells transfected with ANO1 TSB or a negative control ($n = 4$). **e** Migration rates during repair of primary CF cells. Representative photographs were taken during 4 h of wound closure of CF cells (*left*), and migration rates were quantification (*right*) ($n = 4$). *Scale bar* 20 μm. Histograms represent average values ± SDs and were compared using Student's $t$-test. **f** Effect of TSB control or ANO1 TSB on mucus dynamics after 30 days of transfection. The movements of 100 beads were quantified for each condition, and the average speed (μm/ms) was determined. *Scale bar* 40 μm. Histograms represent the mean values ± SDs and were compared using Student's $t$-test

Fig. 19). The chloride efflux was significantly increased (38%) in CF mice treated with ANO1 TSB as compared to CF control mice or WT mice (Fig. 7a). Moreover, there was no difference in chloride efflux between CF control mice and WT mice; this can be explained by the fact that chloride secretion in mice is mostly related to ANO1[19, 20]. Similar results were obtained with the (6-methoxyquinolinio) acetic acid ethyl ester bromide (MQAE) method (Supplementary Fig. 20). To go further and propose another evidence of in vivo effectiveness of TSB treatment, we have evaluated the effect of ANO1 TSB on mucus expression and on mucus clearance. Mucus accumulation in the airways is a key feature in CF disease and ANO1 could be implicated in mucins secretion[21]. First, we have quantified mucin expression in different models treated or not by ANO1 TSB (CFBE41o-, KM4 cells, CUFI cells, primary human bronchial gland cells). By immunohistology and by qPCR, we have not observed a significant increased of MUC5AC or MUC5B expression (Supplementary Fig. 21). To complete this study on mucus, we have also evaluated mucus dynamics directly on the trachea of the mice. For this, we placed phenol red dye immediately in front of the caudal part of the trachea and observed the progression of the dye to the laryngeal part of the trachea (Fig. 7b and Supplementary Movies 8–11). To show that the observed transport was indeed mediated by ciliated cells, we also applied dye to the laryngeal part of the trachea and observed that the dye did not enter the tracheal lumen as it has been done previously in the litterature[22]. Interestingly, we observed a higher speed of the transport of the phenol red dye in the trachea of CF mice treated with ANO1 TSB (Fig. 7b). Additionally, we studied the mucus dynamics using fluorescent beads on the trachea ex vivo, and the average speed of movement of the beads was faster on the trachea of CF mice who received the ANO1 TSB compared to CF mice treated with the control (Fig. 7c and Supplementary Movies 12 and 13). Based on these results, we propose ANO1 TSB as a candidate drug to treat all CF patients, independently of CFTR mutations.

## Discussion

In this study, we demonstrated that miR-9 was overexpressed in CF bronchial epithelial cells and that it directly downregulated ANO1, a CaCC. In addition, we developed an ANO1 TSB that specifically prevents binding of miR-9 to the 3′UTR of ANO1 mRNA, which we previously showed to be decreased in CF[14]. We were able to increase ANO1 expression and chloride activity in in vitro as well as in vivo CF models (cell lines, primary cells, and mice) to a level similar to that in the similar non-CF models. Our results provide evidence that this approach can be considered as an interesting therapeutic approach in CF.

To identify ANO1 targets, we first analyzed miRNA–target interactions predicted by various computational algorithms and focused on their intersections. Of the four miRNAs identified, miR-19a and miR-19b did not regulate ANO1, while miR-144 regulated ANO1 only indirectly as indicated by preliminary experiments (Supplementary Figs. 1–6). In addition, miR-9 and ANO1 levels of expression were inversely correlated. We validated ANO1 mRNA as a direct target of miR-9 using luciferase assays to demonstrate that the decreased ANO1 expression in CF cells was caused by miR-9-mediated ANO1 repression. However, the mechanisms underlying miR-9 deregulation in CF remain currently unknown.

ANO1 has been considered a potential therapeutic target in CF since its discovery in 2008[9]. Indeed, CF is caused by a deficiency in CFTR, which, like ANO1, is a chloride channel expressed throughout the body. Moreover, others and we have previously shown that, similar to CFTR, ANO1 is involved in other pathways deregulated in CF, such as $HCO_3^-$ secretion, cell migration,

and proliferation[12–14]. Thus, targeting of this channel would be a promising therapeutic strategy for all CF patients, independently of CFTR mutations, unlike the current therapies[3]. Targeting of chloride efflux as a therapeutic strategy in CF has been attempted in the 90s by using uridine-5′-triphosphate (UTP) and analogs, which were shown to be able to stimulate chloride secretion in CF respiratory epithelium[23]. UTP activates a signal by binding to a purinergic receptor to release intracellular calcium and activate undefined CaCCs[24]. In vitro, UTP could restore fluid transport, enhance tracheal mucus viscosity and mucus dynamics, inhibit sodium absorption, and increase airway hydration[25, 26]. These promising results led to the development of two drugs, INS365 and INS37217 (later named Denufosol®; Inspire Pharmaceuticals), that were more resistant to enzymatic degradation than UTP[27]. Denufosol was carried into clinical trials using nebulization for delivery into the airways. After promising results had been obtained in the first clinical trials, an international phase 3 clinical trial was conducted in 466 CF patients[28]. Unfortunately, this trial failed to demonstrate any therapeutic benefit and various hypotheses were raised to explain this outcome. First, the molecular identity of the CaCC was discovered after the phase 3 trial, and the exact effects of Denufosol were only empiric and very transient[29]. It is indeed well known that administration of UTP to the apical surface of the epithelium leads to its binding to the P2Y2 receptor and causes a rapid increase in the cytosolic free calcium concentration associated with a temporary rise in chloride efflux that lasts only a few minutes. Second, the stability of this drug in the context of CF remained problematic, with a very short half-life ranging from 3 h in the nasal epithelium and 17 min in CF lungs to 30 s in the blood because of natural nucleosidases[27, 30]. Third, the key selection criteria for young CF patients (mean age, 14.2 years) with only mild or no appreciable baseline impairment of $FEV_1$ (mean, 92% of predicted) for drug efficacy validation could be discussed. Multiple drugs have indeed demonstrated clinical beneficial endpoints for patients besides $FEV_1$[31, 32]. Finally, while targeting CaCC holds promise, clinical benefits have not yet been achieved; for this reason, we focused on a more objective drug approach by specifically targeting ANO1 in the current study.

Based on knowledge of molecular biology, we developed a specific strategy to increase the expression of a particular target protein by using miRNA. The challenge was to create a highly specific molecule, able to induce a strong ANO1 activity, even in a complex system such as CF. To develop a method more accurate than antagomiR- or RNA sponge-based approaches, which silence miRNA and thus affect all miRNA targets, we used a TSB to block the target site(s) of miRNA(s) in the 3′UTR of ANO1[15]. We used a new RNA-based approach combining different technologies (LNA and phosphorothioates) to enhance stability, cell permeability, and specificity of the RNA tool. Another valuable addition to the miRNA tool by the LNA, a bicyclic high-affinity RNA analog in which the ribose ring is chemically locked in an N-type (C3′-endo) conformation, is that it provides the LNA-modified oligonucleotides with high thermal stability during mRNA target hybridization[33, 34]. We studied ANO1 regulation by miR-9 in CF cell lines and primary cultures obtained from patients homozygous for CFTR-F508del, the most frequent CFTR mutation worldwide. In these cells, by using ANO1 TSB, we succeeded in increasing ANO1 expression and channel activity as well as cell migration by an unknown mechanism. Similar results were obtained previously with CFTR, and the authors have suggested that CFTR participates in airway epithelial wound repair by a mechanism involving anion transport that is coupled to the regulation of lamellipodia protrusion at the leading edge of the cells[35]. Importantly, given the importance of cellular migration in the epithelial repair process, this drug might be able

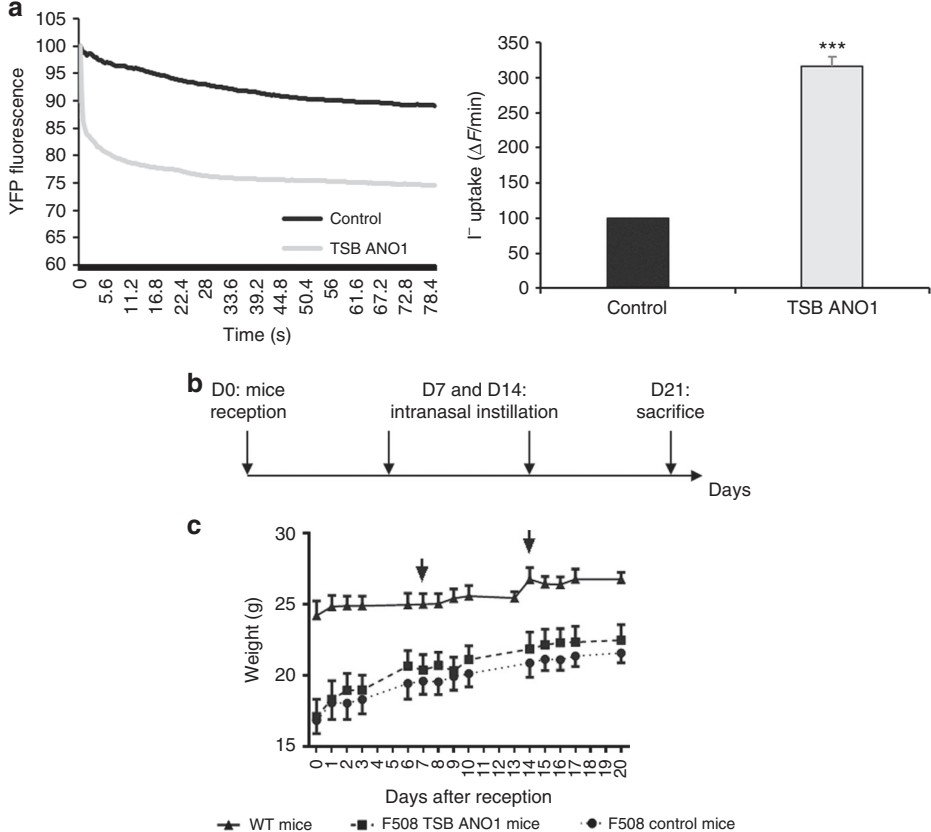

**Fig. 6** miR-9-specific TSB is well tolerated in CF mice. For all the experiments, we used 8-week-old male 129-C*ftr*^tm1Eur CF model mice homozygous for the F508del mutation in the 129/FVB outbred background (F508del-CFTR) and their wild-type littermates obtained from CDTA-CNRS (Orléans, France). **a** Representative and original traces of ANO1 channel activity (*left*) and quantification (*right*) of MLE15 cells transfected with TSB control (control) or ANO1 TSB. Histograms represent average values ± SDs (*n* = 5) and were compared using Student's *t*-test. **b** Outline of the CF mouse experiment. TSB control or ANO1 TSB was instilled intranasally at days 7 and 14 after reception (day 0), and mice were killed at day 21. **c** Growth curves of mice from the day of reception (day 0). *Arrows* represent intranasal instillation

to also enhance bronchial epithelial repair in CF. The therapeutic strategy proposed herein and its mode of action are highly innovative in comparison to other therapies proposed to date. Our approach is independent of the *CFTR* gene mutations, providing a comparative advantage over the recently developed CFTR protein-targeted treatments. Indeed, more than 2,000 *CFTR* gene mutations have been described to date (http://www.cftr2.org); thus, developing a drug that would benefit all patients remains a great challenge[3]. Other challenges, including drug degradation as mentioned above, are at least as important in the context of CF. Artificial miRNA target-site drugs are emerging, and the most promising LNA-based drug currently available is Miravirsen®, an inhibitor of miR-122 currently in clinical trials for the treatment of hepatitis C viral infections[36]. The effects of Miravirsen have been tested in chimpanzees; the LNA-modified anti-miRNA was shown to be accurate, non-toxic, and very stable, the drug could still be detected 8 weeks after the end of the treatment[37]. Furthermore, the effects of Miravirsen on HCV RNA levels were prolonged at the end of the treatment[38]. The successes obtained with Miravirsen are highly motivating for the translation of our basic research study to future clinical applications.

We demonstrated that our ANO1 TSB was able to correct the different parameters analyzed in the CF airways, even in a mouse model in which ANO1 expression is predominant, and independently from the CFTR channel[20]. In the mouse trachea, the tissue anatomically closest to human bronchi, we observed an activation of the chloride efflux and the mucus dynamics that lasted seven days after the last instillation. The stability and

physiological activity of our drug are major advantages in this case. Obviously, the method of administration of the drug into the respiratory tract requires optimization before it can be applied to CF patients.

In conclusion, to our knowledge, the current study is the first to propose a realistic alternative therapy for CF that allows to precisely correct an alternative chloride channel, and the first to report a restoration of the chloride efflux, mucus clearance, and cell migration in a CF context by using a TSB targeting the seed region of miR-9 at the ANO1 3′UTR. Furthermore, this compound represents the first description of ANO1 potentiator that can lead to channel activation independent of an election in $Ca^{2+}$ concentration.

Finally, our results highlight that a thorough understanding of the pathophysiology is essential to develop innovative therapeutic strategies.

## Methods

**Bioinformatics analysis of miRNA target genes**. The role of miRNAs in the regulation of ANO1 expression was examined using a computational approach, which predicted the 3′UTR of ANO1 mRNA to contain various seed regions that are recognized by a variety of miRNAs. The putative miRNAs predicted for ANO1 mRNA were identified and compared using the online target prediction algorithms TargetScan (http://targetscan.org), Pictar (http://pictar.mdc-berlin.de), miRDB (http://mirdb.org/miRDB), and miRANDA (http://www.microrna.org). The NCBI and Ensembl genome browsers (http://www.ensembl.org/index.html) provided information on the human ANO1 transcript (NM_018043; ENST00000355303). The miRBase (http://www.mirbase.org) provided information on miR-9 (MI0000466).

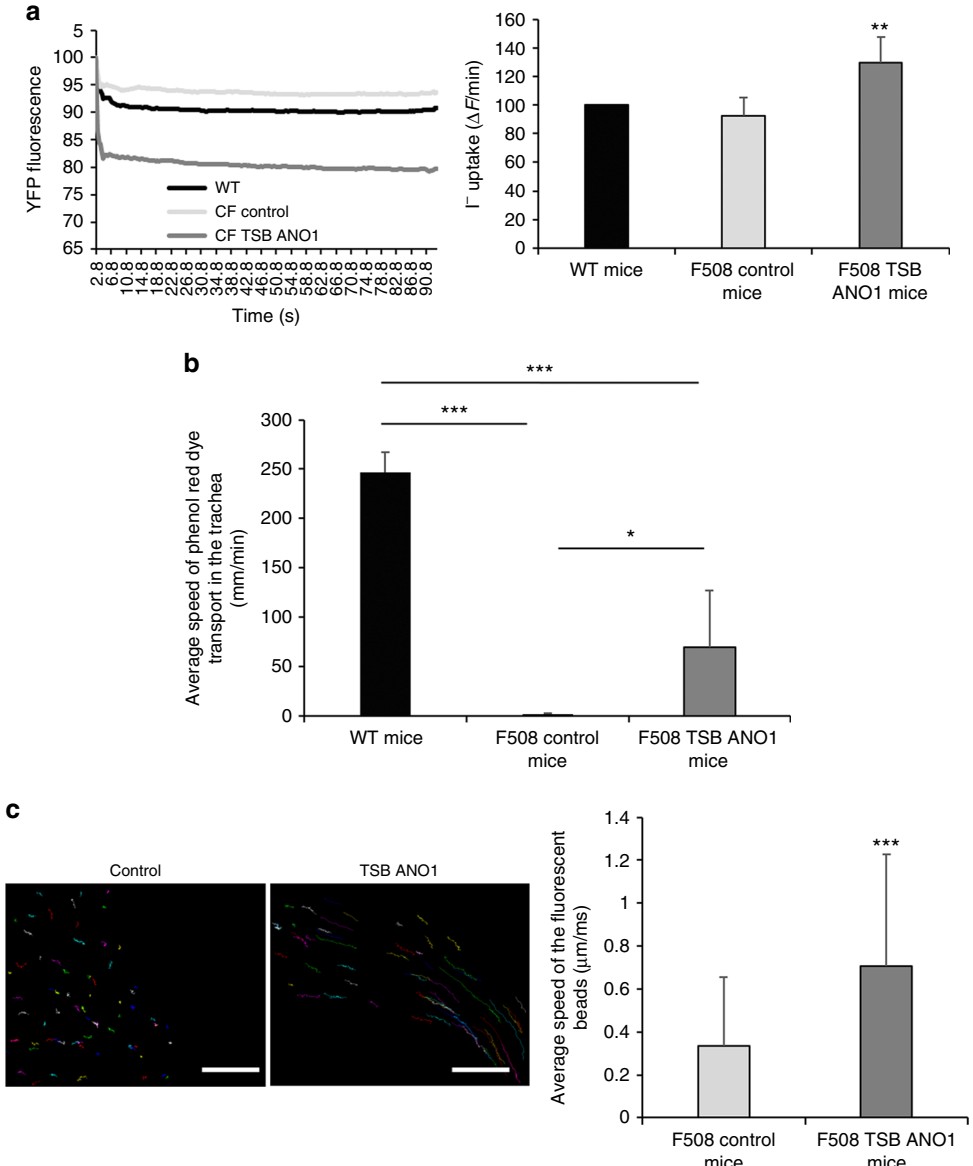

**Fig. 7** miR-9-specific TSB increases ANO1 activity and mucus dynamics in CF mice. For all the experiments, we used 8-week-old male 129-C*ftr*[tm1Eur] CF model mice homozygous for the F508del mutation in the 129/FVB outbred background (F508del-CFTR) and their wild-type littermates obtained from CDTA-CNRS (Orléans, France). **a** Representative and original traces of ANO1 channel activity (*left*) and quantification (*right*) of cells isolated from wild-type mice ($n = 8$), CF mice instilled with ANO1 TSB ($n = 7$) or a negative control ($n = 7$). Histograms represent the average values ± SDs and were compared using one-ANOVA test coupled with Dunnett's, Bonferroni's, and Tukey's post hoc test. **b** Effect of TSB control or TSB ANO1 on the transport of phenol *red dye* in the trachea. Histograms represent the average values ± SDs and were compared using one-way ANOVA test coupled with Dunnett's, Bonferroni's, and Tukey's post hoc test. **c** Effect of TSB control or ANO1 TSB on mucus dynamics on the trachea of CF mice. The movements of 100 beads were quantified for each condition, and the average speed (μm/ms) was determined. *Scale bar* 40 μm. Histograms represent the mean values ± SDs and were compared using Student's *t*-test

**Cell culture**. The human bronchial epithelial cell lines 16HBE14o- (non-CF) and CFBE41o- (CF) were kindly provided by Dr D.C. Gruenert (Institute for Human Genetics, San Francisco, CA, USA) and cultured in MEM containing 10% bovine growth serum and 1% penicillin/streptomycin. Murine lung epithelial cells (MLE-15) and KM4 cell lines were a gift from Dr Caroline Norez (Poitiers-CNRS University, Poitiers, France). MLE-15 were maintained in HITES (RPMI 1640 with 5 μg/ml insulin, 10 μg/ml transferrin, 30 nM sodium selenite, 10 nM hydrocortisone, 10 nM β-estradiol, and 10 mM HEPES) supplemented with 4 mM L-glutamine, 10 % fetal bovine serum, 100 U/ml penicillin G, and 100 μg/ml streptomycin. KM4 cells were grown in Dulbecco's modified Eagle's medium (DMEM) F-12 supplemented with 1% Ultroser G serum substitute (Biosepra,Villeneuve, La Garenne, France), glucose (10 g/l), sodium pyruvate (0.33 g/l), penicillin (100 IU/ml), streptomycin (100 μg/ml), and amphotericin B (2 μg/ml). Cell isolation and subculture procedures of human bronchial gland (HBG) cells were performed on bronchial tissues collected from F508 homozygous CF patients. HBG cells were isolated by enzymatic digestion from bronchial submucosa and grown onto type I collagen-coated 25-cm² tissue culture flasks in a DMEM/Ham's F12 mixture (50/50%, v/v) supplemented with 1% Ultroser G, glucose (10 g/l), and sodium pyruvate (0.33 g/l). Penicillin G (100 U/ml) and streptomycin (100 g/ml) were also added. All media and supplements were obtained from Thermo Fischer Scientific (Courtaboeuf, France) unless otherwise indicated. Cells were maintained at 37 °C in a humidified atmosphere of air with 5% CO₂. All cells were tested for mycoplasma contamination (Lonza, Ambroise, France).

Primary hAECB isolated from bronchial biopsies from CF (F508del/F508del) patients were purchased from Epithelix SARL (Geneva, Switzerland). Fully differentiated ALI cultures (MucilAir,™), were cultured according to the provider's recommendations.

**Cell transfections**. Non-CF and CF cells were transfected with a miR-9 mimic (*mir*Vana® miRNA mimic), miR-9 inhibitor (*mir*Vana® miRNA inhibitor), or negative control (*mir*Vana™ miRNA Mimic, Negative Control) (Thermo Fischer Scientific, Courtaboeuf, France) using HiPerfect (30 nM; Qiagen, Les Ulis, France) according to the manufacturer's instructions. Forty-eight hours after transfection, the cells were lysed for miRNA, RNA, and protein extractions. CF cells were transfected with LNA-enhanced oligonucleotides targeting the miR-9 target site in the ANO1 3′UTR (ANO1 TSB) or with a miRCURY LNA microRNA inhibitor negative control (TSB control) (Exiqon, Denmark) using Interferin (Polyplus, Ozyme, France). Twenty-four hours after the transfection, the cells were lysed for protein extraction or processed for chloride activity or migration assays. hAECB were transfected by adding medium containing LNA control or ANO1 TSB without any transfection reagent to ALI cells. After a 2-h incubation at 37 °C, the medium was removed to restore the ALI condition. Freshly prepared LNA control or ANO1 TSB were added every day for 3 days, and ANO1 expression, chloride activity, and migration were assessed 24 h post treatment.

**Luciferase assay**. For the luciferase assay, we used an ANO1-3′UTR-pMirTarget luciferase plasmid (Origene Technologies, Rockville, USA) and an ANO1-3′UTR-pMir vector-bearing mutations in the miR-9 seed region (MIMAT0000441). CF and non-CF cells were seeded in 24-well plates at $8 \times 10^4$ cells/well and were transfected the next day with 0.5 μg pMir vector and 0.1 μg *Renilla* luciferase vector using Exgen 500 (Euromedex, France). The cells were cotransfected with 30 nM miR-9 mimic, miR-9 inhibitor, or a negative control (Thermo Fischer Scientific, France). Lysates were prepared 48 h after transfection and assayed for both firefly and *Renilla* luciferase using a luciferase assay system (Promega, France). Firefly luciferase activity was normalized to that of *Renilla* luciferase activity.

**RNA and miRNA extraction and qRT-PCR analysis**. RNAs and miRNAs were extracted using a Macherey-Nagel kit (Düren, Germany). miR-9 and RNU6B were reverse-transcribed with the TaqMan MicroRNA Assay kit (Thermo Fischer Scientific) using 20 ng miRNA. ANO1 and GAPDH were reverse-transcribed with the High Capacity cDNA Reverse Transcription kit (Thermo Fischer Scientific) using 1 μg RNA. qPCR was performed using an ABI StepOnePlus (Thermo Fischer Scientific) and TaqMan technology using primers from Thermo Fischer Scientific (Supplementary Table 1). For relative quantification, the ANO1 mRNA level, calculated using the $2^{-\Delta\Delta Ct}$ method, was normalized to the GAPDH mRNA level, and the expression levels of non-CF models and miR-9 were normalized to the RNU6B mRNA level. Each sample was assessed in triplicate.

**Western blotting**. Total proteins were extracted in lysis buffer (150 mM NaCl, 20 Mm Tris-HCl, 2 mM EDTA, 1% Nonidet P-40, 0.1% SDS and inhibitor proteases) on ice for 1 h and then centrifuged for 15 min at 12,000×*g* and 4 °C. The supernatant containing total protein was recovered, and the protein concentration was evaluated using the Protein Assay Kit (Thermo Fisher Scientific, Strasbourg, France) and quantified by spectrophotometry at 450 nm.

Total protein extract (20 μg) was reduced and size-separated by 8% SDS-PAGE and transferred to PVDF membranes (Bio-Rad, Marnes-la-Coquette, France), which were blocked in 5% BSA (PAA, Les Mureaux, France). Then, the membranes were incubated with specific primary rabbit polyclonal antibodies against ANO1 (ab53213, 1:10; Abcam, Paris, France) and mouse monoclonal β-actin (A5441, 1:500; Sigma, Saint Quentin Fallavier, France). The proteins of interest were detected using ECL detection system (Thermo Fisher Scientific). Relative quantification was performed by densitometric analysis using MultiGauge software (Fujifilm, Courbevoie, France). See Supplementary Fig. 22 for gel source data.

**Migration assays**. Migration assays were performed using specific wound assay chambers (Ibidi®; Biovalley, Marne La Vallée, France) that provide uniform wounds between two monolayers. CFBE41o- cells ($3.5 \times 10^4$) were seeded in Ibidi silicone culture-inserts and were incubated at 37 °C with 5% CO₂. After 24 h, the cells were transfected with a miR-9 mimic, negative control (Life Technologies, Saint Aubin, France), or ANO1 TSB/LNA control (Exiqon). The culture inserts were removed after 48 h, leaving a cell-free gap (or wound) for cell colonization. For hAECB, wounds were generated using a tip soaked in liquid nitrogen. Wound closure was observed for 4 h under an Axiovert 200 microscope in a chamber maintained at 37 °C with 5% CO₂. Mean migration rates during wound closure were assessed in three areas of the gap. At each time point and in each field, five lengths were measured using the AxioVision Rel software (Zeiss, Marly-Le-Roi, France).

**ANO1 chloride channel activity assay**. ANO1 channel activity was assessed by $I^-$ quenching of halide-sensitive YFP-H148Q/I152L protein (Thermo Fischer Scientific). The probe was transfected into the cells, and after 48 h of culture, conductance was stimulated with UTP (10 μM). $I^-$ solution (140 mM) was added, and the fluorescence was recorded using a plate reader as previously described[39]. The initial $I^-$ influx rate upon addition of each solution was computed from changes in YFP fluorescence data using non-linear regression. For quantitative analysis, the slope for fluorescence quenching, which correlates to the level of chloride

conductance ($I^-$ uptake), was determined using linear regression. The rate of change ($\Delta F$/min) was then calculated.

**Microinjection experiments**. CF cells were seeded on glass-bottom Ibidi dishes 24 h after transfection with the YFP-H148Q/I152L protein construct, and ANO1 TSB or control TSB were microinjected using a Xenowork micromanipulator and digital microinjector (Sutter Instrument, CA, USA). To discriminate the different transfections, ANO1 TSB was microinjected with Dextran Texas Red neutral (Thermo Fischer Scientific). Four hours after microinjection, the cells were injected with $I^-$ using a microperfusion system (Valvelink; Automate Scientific, CA, USA) and observed under an Axiovert 200 microscope (63 × objective lens, Zeiss). The fluorescence was quantified with the ImageJ software (US National Institutes of Health, ML, USA).

**Mucus clearance assay**. Thirty days after transfection of hAECB cells with ANO1 TSB or control TSB, FluoSpheres Carboxylate-Modified Microspheres, 1.0 μm, yellow-green fluorescent (505/515) (Thermo Fisher Scientific) diluted in culture medium (1/50) were added to the apical face of the cultures. Movement of the beads was recorded under an Axiovert 200 microscope (Zeiss). Fluorescence images were collected every 3 ms and composed into a time-lapse image series using Axiovision software (version 4.6).

**Animal experiments**. All experiments were approved and conducted in accordance with our Institutional Animal Care and Use Department (approval 20150511145844v3 of the Ethical Committee for Laboratory Animal Care Charles Darwin France). For all the experiments, we used 8-week-old male 129-Cftr^tm1Eur CF model mice homozygous for the F508del mutation in the 129/FVB outbred background (F508del-CFTR) and their wild-type littermates obtained from CDTA-CNRS (Orléans, France). Animals were maintained in the specific pathogen-free mouse facility of Paris 06 with access to food and water *ad libitum*, and their weights were recorded every day. To minimize bowel obstruction, a commercially available osmotic laxative containing polyethylene glycol (PEG-3350) and electrolytes (Movicol®) was supplied continuously at 6% in the drinking water[40]. Prior to instillation, mice were anesthetized with 2.5% isoflurane delivered in O₂ (2 l/min). Challenge doses of TSB (10 mg/kg; once per week during 2 weeks) were intranasally administered by pipetting the solution onto the outer edge of each nostril. Seven days after the last challenge, the mice were killed with a lethal dose of CO₂. After exsanguination, the chest wall was opened, and the trachea was isolated. Halide sensor method was performed on a tracheal section, and chloride activity was measured as previously described[14]. For the in situ experiments, mice were immersed in PBS previously heated. Tracheae were washed with PBS, and ~3 μl of phenol red dye was applied to the laryngeal opening. For ex vivo experiments, tracheae were explanted and transferred to culture dishes coated with agarose and were held in position by needles. After covering the trachea with PBS, phenol red dye was applied to the laryngeal part of the trachea.

**Statistical analysis**. XLSTAT version 2014 (Paris, France) was used for all analyses. No statistical method was used to predetermine sample size. Animals were randomly assigned to experimental groups. All analyses were performed in a blinded manner, and all data are presented as the mean ± SD. Outliers were identified by Grubbs's test (http://graphpad.com/quickcalcs/Grubbs1.cfm). Statistically significant differences between two groups were determined by two-tailed *t*-test. An unpaired Student's *t*-test with Welch's correction was applied when variances were not equal. For three or more groups, statistical analysis was performed using one-way analysis of variance (ANOVA), followed by the Bonferroni, Tukey, and Dunnett post analysis, as appropriate; $P < 0.05$ was considered significant. Correlation coefficients were calculated by Pearson's correlation test. In the figures, statistically significant differences with *$P < 0.05$, **$P < 0.01$, and ***$P < 0.001$ are indicated.

**Data availability**. The data that support the findings of this study are available from the corresponding author upon reasonable request.

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

## Acknowledgements

We thank O. Bregerie and coworkers of the Animal Research Facility of the Pitié-Salpétrière-UPMC Université Paris 06, France. We thank Dr D.C. Gruenert for providing the 16HBE14o- and CFBE41o- cells and Dr C. Norez for providing MLE-15 cells. This research was funded in part by grants from Inserm, UPMC-Paris 06, the non-profit organization Vaincre la mucoviscidose, and the Legs Poix-Chancellerie des Universités, Paris. F.S. received a Ph.D. grant from Vaincre la Mucoviscidose. M.R. received a Ph.D. grant from the Émergence-UPMC 2010 research program.

## Author contributions

F.S. performed and designed experiments, interpreted results, and wrote the manuscript. M.R. performed experiments. C.C. provided human ALI culture. N.R. performed experiments. P.L.R. interpreted experiments and wrote the article. S.B.-L. provided human ALI culture and performed experiments. H.C. interpreted data and wrote the article. O.T. performed and designed experiments, interpreted results, obtained funding, and wrote the manuscript.

## Additional information

**Competing interests:** F.S. and O.T. own a patent application related to the findings described herein (PCT/FR2015/051850). The remaining authors declare no competing financial interests.

