## [Peer Review file · Nature Communications]

Reviewers' comments:

Reviewer #1 (Remarks to the Author):

1. The paper contains interesting new information, including a potential therapeutic option, and is generally well written.
2. I was expecting to see a number of experiments which I struggled to find. These included:
 - a) the effect of a miR-9 mimic on CF cells (to match the non-CF data presented)
 - b) the effect of a miR-9 inhibitor on non-CF cells (to match the CF data presented).
3. The authors need to be careful about the description of the ANO1 TSB data at the end of Results and the start of Discussion. There is no non-CF versus CF difference in the mice. The TSB elevates chloride to supra-normal levels. This should be clearly indicated and discussed in terms of any potential downsides.
4. There were a number of places where I would urge the authors to tone down the text. For example the last line of the Abstract 'holds great promise'; it is hard to conclude this given the current data. Similarly on P4, 'remarkable increases' etc.

Reviewer #2 (Remarks to the Author):

The m/s by Sonnevile et al. reports on the effect of a LNA-based TSB on reversal of miR-9 mediated ANO1/TMEM16A repression in CF bronchial epithelial cells. The authors demonstrate that; (i) miR-9 is higher and ANO1 is lower in a CF versus a non-CF BEC line; (ii) miR-9 overexpression decreases ANO1 mRNA, protein and channel activity, and decreases cell migration in non-CF cells; (iii) miR-9 directly regulates the ANO1 3'UTR in CFBEs; (iv) an ANO1-specific miR-9 blocking TSB increases ANO1 channel activity after transfection into CFBEs (measured by 2 methods) and cell migration rate; (v) the TSB increases ANO1 protein expression, channel activity and mucus transport when applied to primary ALI F508del homozygous BECs; (vi) the TSB increases ANO1 activity in a wildtype mouse BEC cell line; and (vii) intranasal administration of the TSB to F508delCFTR homozygous mice led to no weight loss and increased chloride efflux from their dissected tracheas. The authors conclude that the TSB has therapeutic potential to correct chloride ion efflux in people with CF.

This paper will be of strong interest to researchers in the fields of CF, electrophysiology and miRNA and is in line with current thinking in these fields. Although there is a full story presented more data for some individual aspects of the work should be provided. The work uses a similar approach as Viart et al. ERJ 2015 however the miRNAs and channels under investigation are different; the approach here is novel in that it has therapeutic potential for any CFTR genotype.

The paper is of an appropriate length and is well written with some minor clarifications required on certain niche concepts for a non-specialist reader. Other CF miRNA studies are referred to by citing only one review article written by this group. The methods are generally clear and reproducible, apart from the too brief description and no supporting citation for the mucus clearance assay. P values are not provided. Animal studies were performed according to ethical guidelines and human primary CF cells were purchased.

1. The paper refers to CF cells/cell lines throughout however only one human CF cell line was tested in Figures 1-4. Are miR-9/ANO1 expression levels increased/decreased in other CF versus non-CF bronchial epithelial cell lines? Do miR-9/ANO1 show reciprocal expression patterns in the primary CF cells?
2. The relevance of the wound closure assay, as used here, is not entirely clear. Is it an appropriate read-out for TSB function or is it an epiphenomenon? What happens after the 4h wound healing period - is proliferation also affected? This could be measured at later time-points and using other assays - increased proliferation would not be a desirable effect. How does ANO1

enhance epithelial cell migration? Do the authors have more convincing images than those currently shown in figures 2d and 4d?

3. Although a video of TSB microinjection is provided as supplementary material and referred to on page 4 line 188, Figure 4b is (possibly) the only figure that used this approach. Please clarify.

4. In figure 5a, the depiction of 1 cell per view is insufficient.

5. The authors interchangeably refer to 'ciliary beat frequency' and 'mucus dynamics' (but not in the relevant results section heading or title of figure legend 5). These are two different processes. What the authors are measuring here is movement of mucus which can be affected by a number of factors including hydration. CBF assays visualise actual beating of cilia and uses a different method than described here.

6. The measurement of body weight in the animal study is not an adequate measure of toxic side effects. Inflammatory cell profiles and cytokine assays of nasal lavage, for example, would provide much clearer evidence of local toxic or inflammatory effects or the lack thereof.

7. Regarding the supplementary data: the scale bar is not visible on Sup Fig 4, the legend for Sup Fig 5 does not clearly explain the figure, in Sup Fig 7 and 8 only human and murine sequences should be shown for clarity.

Reviewer #3 (Remarks to the Author):

This manuscript presents a very interesting therapy for cystic fibrosis. The data show, that miRNA-9 is upregulated in CF and downregulates ANO1 expression. Blockage of miR-9 using a microRNA target site blocker (TSB) increases ANO1 activity and thus compensate the CFTR deficiency.

However, there are basic problems:

The authors claimed that ANO1 is expressed in the native lung epithelium. Nevertheless, there are publications, also cited by the authors, showing that ANO1 is only expressed in human lung cancer cells, but not in healthy lung cells (Jia, L. et al. PLoS One 429 10, (2015). Huang, F. et al. (PNAS 109, 16354, 2012) found that ANO1 is upregulated in inflamed tissue using an asthma model.

Therefore, it is necessary to show in the mouse model in which lung cells ANO1 is expressed; it is also necessary to show the downregulation of ANO1 expression in native CF tissue and the upregulation in the native tissue after treatment of the animal with TSB, too. As mentioned by the authors, ANO1 is promoting proliferation and migration of cells. Cultured cells or primary isolated cells may express ANO1 due to dedifferentiation processes under culture conditions.

Suppl. Fig.4 shows ANO1 expression in non-CF cells. Surprisingly, ANO1 is expressed in cytosolic membranes, but not in the plasma membrane.

The I- quenching assay (Fig. 2, Fig. 5) is not specific for ANO1. Which other Cl- channels are expressed in the used cell lines? Do the cell lines express other anoctamins and could they be affected by miR-9 as well?

Fig. 2 is showing the migration rate after transfection with mimic miR-9 of non-CF cells. Is the migration rate also reduced in CF-cells?

The ciliary beat frequency was investigated in human bronchial epithelial cells, but there was no link to a figure showing the data of the speed increase in ANO1 TSB expressing cells. As mentioned above, it makes more sense to measure mucociliary clearance directly in the treated mice.

Mimic miR.9 caused a 40% reduction of luciferase gene expression from WT-ANO1 3'UTR as compared to mut-ANO1 3'UTR. Why is this effect weaker compared to the downregulation of ANO1 in CF cells?

Minor points:

Introduction: The statement ANO1 is involved in HCO₃⁻ permeability is without citation.

Mimic miR-9 or mir-9 mimic?

Response to Reviewer #1:

1. The paper contains interesting new information, including a potential therapeutic option, and is generally well written.

2. I was expecting to see a number of experiments which I struggled to find. These included:
a) the effect of a miR-9 mimic on CF cells (to match the non-CF data presented)
b) the effect of a miR-9 inhibitor on non-CF cells (to match the CF data presented).

Response: In response to the reviewer's apt suggestion, we have conducted these experiments; the results are provided below. The protocol of transfection is described in the manuscript (Fig. 2 & Fig. 3). We have included these results in the revised version of the manuscript (Supplementary Figs. 3) and in the revised manuscript. CF cells transfected with a mimic of miR-9 exhibit a significant increase of luciferase-3'UTR ANO1 activity and non-CF cells transfected with an inhibitor of miR-9 exhibit a significant decrease (lines 111-114, pages 3-4).

Supplementary Figure 3 Relative luciferase activity in CF and in non-CF bronchial epithelial cells (16HBE14o-) transiently transfected with a luciferase-3'UTR ANO1 vector with modulators of miR-9. Relative luciferase activity in CF cells (CFBE41o-) (a) and in non CF cells (16HBE14o-) (b) transiently transfected with luciferase-3'UTR ANO1 and cotransfected with an inhibitor (inh miR-9) or a mimic of miR-9. Firefly luciferase activity was normalized to Renilla luciferase activity. Histograms represent average values \pm SDs and were compared using Student's t-test (n = 3 with 8 replicates).

3. The authors need to be careful about the description of the ANO1 TSB data at the end of Results and the start of Discussion.

Response: We agree with the reviewer; we have revised the description of the ANO1 TSB data in Results and Discussion sections.

There is no non-CF versus CF difference in the mice. The TSB elevates chloride to supra-normal levels. This should be clearly indicated and discussed in terms of any potential downsides.

Response: We agree with the reviewer. We have clearly stated that there is no difference in ANO1 chloride efflux between CF and non-CF mice and we have discussed this point in the new version of the text (lines 179-181; page 5).

4. There were a number of places where I would urge the authors to tone down the text. For example, the last line of the Abstract 'holds great promise'; it is hard to conclude this given the current data. Similarly on P4, 'remarkable increases' etc.

Response: In agreement with the reviewer's comment, we toned down our claims at the appropriate instances (line 31, p. 1; lines 129 & 131, page 4).

Response to Reviewer #2:

The m/s by Sonnevile et al. reports on the effect of an LNA-based TSB on the reversal of miR-9 mediated ANO1/TMEM16A repression in CF bronchial epithelial cells. The authors demonstrate that; (i) miR-9 is higher and ANO1 is lower in a CF versus a non-CF BEC line; (ii) miR-9 overexpression decreases ANO1 mRNA, protein and channel activity, and decreases cell migration in non-CF cells; (iii) miR-9 directly regulates the ANO1 3'UTR in CFBEs; (iv) an ANO1-specific miR-9 blocking TSB increases ANO1 channel activity after transfection into CFBEs (measured by 2 methods) and cell migration rate; (v) the TSB increases ANO1 protein expression, channel activity and mucus transport when applied to primary ALI F508del homozygous BECs; (vi) the TSB increases ANO1 activity in a wildtype mouse BEC cell line; and (vii) intranasal administration of the TSB to F508delCFTR homozygous mice led to no weight loss and increased chloride efflux from their dissected tracheas. The authors conclude that the TSB has therapeutic potential to correct chloride ion efflux in people with CF.

This paper will be of strong interest to researchers in the fields of CF, electrophysiology, and miRNA and is in line with current thinking in these fields. Although there is a full story presented more data for some individual aspects of the work should be provided. The work uses a similar approach as Viart et al. ERJ 2015 however the miRNAs and channels under investigation are different; the approach here is novel in that it has therapeutic potential for any CFTR genotype. The paper is of an appropriate length and is well written with some minor clarifications required on certain niche concepts for a non-specialist reader. Other CF miRNA studies are referred to by citing only one review article written by this group. The methods are generally clear and reproducible, apart from the too brief description and no supporting citation for the mucus clearance assay. P values are not provided. Animal studies were performed according to ethical guidelines and human primary CF cells were purchased.

1. The paper refers to CF cells/cell lines throughout however only one human CF cell line was tested in Figures 1-4. Are miR-9/ANO1 expression levels increased/decreased in other CF versus non-CF bronchial epithelial cell lines? Do miR-9/ANO1 show reciprocal expression patterns in the primary CF cells?

Response: We thank the reviewer for this useful comment. We analyzed miR-9 and ANO1 expression in different cell models to have more convincing data. We have assayed the expression of ANO1 and miR-9 in different matched cells published (CUFI/NULI; Saint-Criq V. et al., PloS One, 2013) in primary cells (bronchial glandular cells (Tabary O et al., Am J Pathol, 1998)) and cells cultivated in an air-liquid interface. In all the models tested, the results clearly indicated an inverse correlation between ANO1 expression and miR-9 expression. For example, in human bronchial primary cells, we observed a significant correlation between miR-9 and ANO1 expression (n=3 in triplicate; P<0.026). In the ALI cell culture from CF patients (n=5 F508del/F508del), in our opinion the best model used in our study, we observed a significant correlation (P<0.003) between overexpression of miR-9 and low expression of ANO1. We have mentioned these results in the revised version of the manuscript (lines 74-77, page 3). To keep the main message clear, we have not included the data in the manuscript.

2. The relevance of the wound closure assay, as used here, is not entirely clear. Is it an appropriate read-out for TSB function or is it an epiphenomenon? What happens after the 4h wound healing period - is proliferation also affected? This could be measured at later time-points and using other assays – increased proliferation would not be a desirable effect.

Response: We appreciate this comment; however, as previously published (Ruffin et al. *Biochim Biophys Acta*, 2013), we focused on the wound closure assay on migration independently of cell proliferation, which is why we included only this result until 4h. As previously shown, ANO1 modulation, as CFTR, could have an effect on proliferation by an unknown mechanism (Jia L. et al., *PLoS One*, 2015). So, we decided to focus the analysis only on cell migration until 4h after wound healing. Therefore, to address the reviewer's concern, we have performed proliferation assays of CF cells in wound closure assays until 8 h. The results exhibited significant proliferation occurred after 4 h, indicating that at this time point, proliferation could have a substantial role and interfere with wound closure assay. Thus, we have decided to maintain the initial data limited to 4 h of wound healing. We have mentioned these results in the revised paper (data not shown; lines 94-96, page 3).

Cell proliferation index of cells during repair. At $t=0$, a wound was generated in CFBE41o- cell culture and proliferation index was estimated by Cytoquant NF cell proliferation assay kit (ThermoFischer) following manufacturer's instruction. Proliferation index was normalized to the control at $t=0$. Histograms represent average values \pm SDs and were compared using Student's t-test ($n = 3$ in duplicate).

How does ANO1 enhance epithelial cell migration?

Response: We would like to thank the reviewer for this comment. Unfortunately, to our knowledge, the mechanism is not described. Jia et al. have demonstrated that ANO1 overexpression is associated with lung cancer, but the role of this protein remains unknown (Jia et al. *PLoS One*, 2015). Similarly, Guan L et al. clearly demonstrated that the inhibition of ANO1 suppresses proliferation by an unknown mechanism (Guan L et al. *Oncotarget*, 2016). Similar results were obtained previously with CFTR, and the authors have suggested that CFTR participates in airway epithelial wound repair by a mechanism involving anion transport that is coupled to the regulation of lamellipodia protrusion at the leading edge of the cells (Schiller et al., 2010). We have mentioned these findings in the revised manuscript (lines 250-253, page 6).

Do the authors have more convincing images than those currently shown in figures 2d and 4d?

Response: Unfortunately, we do not have more convincing images at 4 h; which is why we included quantification data in Figures 2 and 4. As indicated previously (Ruffin et al., 2013), we decided to analyze on migration only. Additional experiments revealed a significant increase in proliferation 4 h after initiation of the wound healing process. In figure 4, the contrast is limited by the filter in the ALI cultures. In the revised manuscript, we have added red lines in Figures 2d and 4d to clearly indicate the edges of wound healing.

3. Although a video of TSB microinjection is provided as supplementary material and referred to on page 4 line 188, Figure 4b is (possibly) the only figure that used this approach. Please clarify.

Response: We are sorry for this mistake, and we have corrected it. Indeed Figure 4b is the only figure for which microinjection experiment was used.

4. In figure 5a, the depiction of 1 cell per view is insufficient.

Response: In the revised manuscript version, we have included s data obtained by a confocal microscopy of CF cells cultured in an air-liquid interface (Figure 5a; see image below). Moreover, we have added a Supplemental Video 3 showing a 3D visualization of fluorescent TSB in the cells.

Figure 5a: Confocal microscopic analysis of fluorescein-conjugated TSB transfected into human bronchial cells isolated from CF patients (green). Cells were cultured in ALI and transfected for 24 h and recorded with Axiovert 200 microscope (Zeiss). The different images were computed with ImageJ.

5. The authors interchangeably refer to ‘ciliary beat frequency’ and ‘mucus dynamics’ (but not in the relevant results section heading or title of figure legend 5). These are two different processes. What the authors are measuring here is movement of mucus which can be affected by a number of factors including hydration. CBF assays visualize actual beating of cilia and uses a different method than described here.

Response: We apologize for this lapse. In agreement with the reviewer, we have modified the text accordingly, we indeed performed mucus dynamics experiments (line 153, page 4 & line 215, page 6).

6. The measurement of body weight in the animal study is not an adequate measure of toxic side effects. Inflammatory cell profiles and cytokine assays of nasal lavage, for example, would provide much clearer evidence of local toxic or inflammatory effects or the lack thereof.

Response: We thank the reviewer for this insightful comment. Previously, different groups using (target site blocker) TSB with the same technology have addressed this question. For example, Roberts et al. have demonstrated that LNA oligonucleotides have little or no effect in the heart, lungs, and kidneys after 4 daily intraperitoneal injections in mice (Roberts. J et al. Mil Ther 2006). In another model, Hildebrandt-Eriksen, et al. intravenously treated 10 cynomolgus monkeys with various dose of an LNA TSB (Miravirsen) weekly for 4 weeks and the authors concluded that the maximal inhibition obtained with the drug was associated with limited changes, indicating that clinical assessment of the drug is warranted (Hildebrandt-Eriksen et al. Nucleic Acid Ther, 2012). Moreover, patients treated with Miravirsen showed no dose-limiting adverse effects after 5 weekly subcutaneous injections during a 29-day period (Jansenn H et al., N Engl J Med, 2013). Of course, the sequence and target are completely different; thus, we have analyzed the toxicity in our model with our specific drug. For a more rigorous assessment, we have decided to work on the same samples used in the previous version, 7 days after the last instillation of the TSB. We have used the official recommendation of the American Thoracic Society for the measurements of experimental acute lung injury in animals (Matute-Bello et al. Am J Respir Cell Mol BiolAm J Respir Cell Mol Biol, 2011). Following these recommendations, we have analyzed cytokine (IL1 β , IL6, and KC) mRNA expression using quantitative PCR in the trachea and lungs. The results clearly indicated that TSB has no significant effects on inflammation *in vivo* 7 days after

the last intranasal administration. We have included these results (data not shown) in the revised version of the manuscript (lines 174-179, page 5). The results are showed below.

Expression of mRNA (IL1β, KC, and IL-6) in the trachea of CF mice (F508del/F508del) treated with or without ANO1 TSB. Expression of mRNA of IL1β, KC, and IL-6 was assayed by RT-qPCR of CF mice treated with ANO1 TSB or negative control. TSB control or ANO1 TSB was instilled intranasally at days 7, and 14 after reception and mice were sacrificed at day 21.

7. Regarding the supplementary data: the scale bar is not visible on Sup Fig 4, the legend for Sup Fig 5 does not clearly explain the figure, in Sup Fig 7 and 8 only human and murine sequences should be shown for clarity.

Response: We thank with the reviewer for pointing these out, we have modified the Sup Fig 5, Sup Fig 6, Sup Fig 8 and Sup Fig 9 in the revised version of the manuscript.

Supplemental Fig. 8 & 9: Sequence alignment of miR-9 and near the seed region of miR-9 on ANO1 3'UTR.

Response to Reviewer #3:

This manuscript presents a very interesting therapy for cystic fibrosis. The data show that miRNA-9 is upregulated in CF and downregulates ANO1 expression. Blockage of miR-9 using a microRNA target site blocker (TSB) increases ANO1 activity and thus compensate the CFTR deficiency. However, there are basic problems:

1. The authors claimed that ANO1 is expressed in the native lung epithelium. Nevertheless, there are publications, also cited by the authors, showing that ANO1 is only expressed in human lung cancer cells, but not in healthy lung cells (Jia, L. et al. PLoS One 429 10, (2015). Huang, F. et al. (PNAS 109, 16354, 2012) found that ANO1 is upregulated in inflamed tissue using an asthma model. Therefore, it is necessary to show in the mouse model in which lung cells ANO1 is expressed. It is also necessary to show the downregulation of ANO1 expression in native CF tissue and the upregulation in the native tissue after treatment of the animal with TSB, too.

Response: In the revised manuscript, we have added the references pointed out by the reviewer (10, 13, and 18; Jia et al., PloS One, 2015; Huang et al., PNAS, 2012; and Rock et al., Dev Biol,

2008, respectively). We highly appreciate this critical comment. As previously shown the effects of TSB on mRNA expression are very weak, the different mechanisms are reviewed in Valinezhad-Orang et al. (Valinezhad Orang, A., R. Int J Genomics, 2014). In fact, in physiological conditions, the fixation of TSB does not induce mRNA degradation in all case but can induce a decrease in protein processing and/or in protein activity. As for the first part of the question, we added results on ANO1 mRNA quantification (data not shown) in the revised manuscript (lines 180 & 181, page 5); no significant difference was found in CF mice treated with ANO1 TSB as compared to CF control mice or WT mice.

Unfortunately, because of limited tissues quantity, we were not able to measure ANO1 protein expression analysis. Therefore, we have focused on ANO1 activity rather than ANO1 expression (see below). As for the second part of the question, unfortunately, our model is limited by the absence of lung disease development in F508-del CFTR mice. CF lung pathology has been studied intensively using CF mutant mice and has been thoroughly reviewed (Wilke M, J Cyst Fibros 2011; Lavelle GM, Biomed Res Int, 2016). No CF mouse model developed spontaneous lung inflammation without a challenge. The lack of severe spontaneous lung pathology has been partially attributed to the expression of a non-CFTR calcium-activated channel and the fact that this expression serves to rectify the ion imbalance underlying CF lung disease. Moreover, because of differences in cellular architecture, physiology, host-pathogen interaction, lifestyle, and lifespan, it is not surprising that CF mice do not reproduce exactly the lung disease typical of CF patients. Unchallenged CF mutant mice present a modest increase in the number of mucus-producing cells in the proximal airways and mucus hypersecretion in lavage. However, overt airway plugging and air trapping such as that reported in CF patients and recently in CF ferrets and pigs are not observed. Thus, in this context, the CF model and the route of administration pose limits for us to adequately respond to the question.

Therefore, we used chloride efflux rather than protein or mRNA expression as the primary parameter. Previous results in the CFTR field have demonstrated that CFTR expression in some cases does not sufficiently correlate with chloride activity. Moreover, one report has clearly shown that ANO1 activity is determined by alternative splicing (Ferrera et al., J Biol Chem, 2009) and/or by submicromolar calcium concentrations (Eggermon et al., Proc Am Thorac Soc, 2004) that could explain a difference between ANO1 expression and ANO1 activity.

To be able to present more convincing results, we have quantified 2 different chloride effluxes during the experiment using the halide sensor method and the more classical MQAE method; the results are presented below. The results obtained with MQAE are similar to those obtained by the halide sensor method. As previously demonstrated, chloride secretion in CF mouse trachea is mainly independent of CFTR (Gianotti et al. J Cyst Fibros, 2016) in CF mice; and mainly due to ANO1 channel (Rock JR. et al., J Biol Chem, 2009). We have added this new result in the revised article (data not shown; lines 181-183 page 5).

Quantification of trachea isolated from wild-type mice (n=8), CF mice instilled with ANO1 TSB (n=7) or a negative control (n=7). The tracheas were incubated with MQAE during 30 minutes before analysis. Histograms represent the average values \pm SDs and were compared using one-ANOVA test coupled with Dunnett's, Bonferroni's and Tukey's posthoc test.

2. As mentioned by the authors, ANO1 is promoting proliferation and migration of cells. Cultured cells or isolated primary cells may express ANO1 due to dedifferentiation processes under culture conditions.

Suppl. Fig.4 shows ANO1 expression in non-CF cells. Surprisingly, ANO1 is expressed in cytosolic membranes, but not in the plasma membrane.

Response: We thank the reviewer for this interesting comment. We have performed this experiment in non-polarized cells; in this case, it is very difficult to observe localization at the surface of the cells. For this reason, we have deleted the Supplementary Figure 4 and have focused on chloride activity experiments.

3. The I- quenching assay (Fig. 2, Fig. 5) is not specific for ANO1. Which other Cl⁻ channels are expressed in the used cell lines? Do the cell lines express other anoctamins and could they be affected by miR-9 as well?

Response: We agree with the reviewer that the Premo halide sensor is not specific for ANO1 activity. In a previous experiment (Ruffin et al., *Biochim Biophys Acta*, 2013), we have demonstrated that a specific siRNA against ANO1 completely inhibited UTP chloride activity in CFBE41o- cells.

In response to this comment, we have included new results obtained in CFBE cells stable ANO1 KO using a shRNA. In this case, we observed a very strong inhibition of UTP-dependent chloride efflux. We have included this result in the revised version of the manuscript (lines 134-135, page 4; and added the data to Supplemental Fig. 7).

Supplementary Figure 7 Validation of ANO1 chloride efflux by halide sensor method.

CF cells (CFBE41o-) were stably transfected with shRNA plasmid directed against ANO1 labeled with RFP dye (Origene, Rockville, USA). Cells were selected in culture with puromycin and sorted by cytometry. Chloride activity was assayed by the Premo halide sensor method.

In another project, we conducted a transcriptomic analysis of CF cells cultured in an air-liquid interface. In this case, in all patients, the number of reads (representative of mRNA expression) was insignificant for ANO2 (<50 reads per patient), bestrophin 2-4 (<10 reads per patient), and CLCA1-3 (<10 reads per patient). Only bestrophin-1 (BEST1) was significantly expressed (> 2,000 reads per sample); therefore, we carried out a qPCR in ALI cells to observe BEST1 expression under TSB treatment. No significant BEST1 expression is observed under TSB treatment. To date, the cellular role and the expression of this channel are not very clear (Kunzelman K; *Trends Biochem Sci*, 2015), some studies reported BEST1 expression only in the basolateral retinal pigment epithelium cells. Most of the studies suggested that BEST1 may be a Ca²⁺-dependent ion channel by acting as a counterion channel for Ca²⁺ efflux out of the ER. BEST1 may also function in intracellular compartments and help to acidify endosomes. Thus, the contribution of BEST1 in chloride efflux seems to be minor compared to CFTR or ANO1 in our

model as demonstrated by the shRNA experiment. We have clarified this point in the revised version of the manuscript (lines 134-135 page 4).

4. Fig. 2 is showing the migration rate after transfection with mimic miR-9 of non-CF cells. Is the migration rate also reduced in CF-cells?

Response: We thank the reviewer for this remark, this point was investigated in a previous article (Ruffin et al.; *Biochim Biophys Acta*, 2013). For clarity, we have decided to maintain the figure comparing the effects of control and mimic, but we have included this point in the revised version (lines 48-49; page 2).

5. The ciliary beat frequency was investigated in human bronchial epithelial cells, but there was no link to a graph showing the data of the speed increase in ANO1 TSB expressing cells. As mentioned above, it makes more sense to measure mucociliary clearance directly in the treated mice.

Response: During the experiment, we have only analyzed mucus dynamics and not ciliary beat frequency. We have revised the text to avoid any confusion (line 155, page 4 & lines 215-216, page 5). We agree with the reviewer that measuring mucociliary clearance in the treated mice would be more appropriate; however, we have not the expertise required to perform this experiment. A new project to analyze mucus composition and ciliary beat frequency *in vitro* and *ex vivo* induced by ANO1 TSB treatment is on the way.

6. Mimic miR.9 caused a 40% reduction of luciferase gene expression from WT-ANO1 3'UTR as compared to mut-ANO1 3'UTR. Why is this effect weaker compared to the downregulation of ANO1 in CF cells?

Response: We agree with the reviewer that the effect is weaker with the inhibitor of miR-9 than with the mimic. In the literature, many studies have reported this effect of antagomir (Fabbri, E. et al., *Am J Respir Cell Mol Biol*, 2014; Serr I. et al., *Proc Natl Sci U S A*, 2016; Chen L. et al. *Am J Pathol*. 2016, etc.), but the exact mechanism remains unknown.

7. Minor points:

Introduction: The statement ANO1 is involved in HCO₃⁻ permeability is without citation.

Response: In the new version of the text, we have added the appropriate reference 12 (Jung, J., et al. Dynamic modulation of ANO1/TMEM16A HCO₃⁻ permeability by Ca²⁺/calmodulin. *Proc Natl Acad Sci U S A*, 110, 360-365 (2013)).

Mimic miR-9 or mir-9 mimic?

Response: Our editor confirmed the used term to be correct. Mimic is used as a noun in the latter case.

Reviewers' comments:

Reviewer #1 (Remarks to the Author):

The authors have made the appropriate changes in the text and performed additional experiments as requested.

I believe the latter provide the expected reassuring data, but with apologies, am confused about the accompanying text.

Thus:

a) Supplementary Figure 3a shows CF cells. My expectation was that the miR-9 mimic would DECREASE luciferase-3' UTR ANO1 activity, and I believe this is what the figure shows. The accompanying text tells me that there was a significant INCREASE.

b) Supplementary Figure 3b shows non-CF cells. My expectation was that the miR-9 inhibitor would INCREASE ANO1 activity, and I believe this is what the figure shows. The accompanying text tells me there was a significant DECREASE.

Also, the natural place to refer to each component of the new data would be next to the data for the already inserted data. Thus, rather than a sentence added at the end of the paragraph covering both sets of new results, I would suggest the reader would benefit from keeping all mimic data together (CF and non-CF) and all inhibitor data together (CF and non-CF).

Reviewer #2 (Remarks to the Author):

The authors have addressed all of my comments. That said, I would encourage them to mention the proliferation effect observed between 4-8 hours as something that requires attention as this drug is developed further (perhaps around page 6 line 254.)

Reviewer #3 (Remarks to the Author):

The response to the first point of criticism is not satisfying. It is important to know in which cells ANO1 is upregulated by the TSB treatment. The ANO1 activity could be upregulated in the airway epithelial cells or for example in club cells. Increased ANO1 activity in epithelial cells could compensate CFTR deficiency, but upregulation of ANO1 in club cells could produce more mucus like in the asthma model shown before (Huang F et al, 2012). Therefore, it should be checked by immunostaining if TSB treatment increases ANO1 expression on the apical membrane of airway epithelial cells (see Huang et al).

The Iodide/YFP quenching assay measures the Iodide permeability induced by UTP, this is correlated to the increased activity of chloride channels but not to the chloride efflux, mentioned sometimes in the manuscript. Chloride efflux or influx is dependent on the membrane potential and on the intra- and extracellular chloride concentrations, which are not measured by this assay. The data of the YFP quenching assays are confusing: The axis of the graphs are differently labelled: Y axis with and without percentage, X axis with and without time (s). The summaries are not fitting to the original curves, for example in Fig. 6a the value of the control in the summary is 100 dF/min or 1,6 dF/sec, for TBS ANO1 dF300/min or 5 dF/sec. Compared to Fig 6d: WT mice 1,6 dF/sec and F508 TSM ANO1 mice 2 dF/s. The original curves in 6a show for control approximately changes of 1%/s and 15%/s for TSB ANO1 and 6D show for WT mice approximately 5%/sec and for CF TSB ANO1 more than 15 %/s. In Fig 4c the initial rate of Iodide uptake for 16HBE and CFBE TSB ANO1 cells are more or less the same 100 dF/min. The initial slope of the originals on the left side are different. For 16HBE cells approximately 5%/s and for TSB ANO1 15%/s. To avoid confusion: original values, not normalized values, should be shown for the original traces. In

addition, the labels of the curves and of the summary are in some cases different.

The MQAE data shown are not conclusive. The change of intracellular chloride concentration should be shown before and after stimulation with cAMP or UTP.

To claim an alternative strategy to correct chloride efflux in CF patients, *in vivo* data are helpful. At least the effect of the TBS treatment could be easily confirmed by *ex vivo* experiments. Instead of CF cells, the trachea could be used in the same way as described in the manuscript to study the mucus dynamics. For this, the trachea could be mounted in a humidified chamber with basolateral perfusion. Movement of beads could be investigated before and after addition of apical UTP on the trachea from control and TBS treated mice.

Response to Reviewer #1:

Reviewer #1 (Remarks to the Author):

The authors have made the appropriate changes in the text and performed additional experiments as requested.

I believe the latter provide the expected reassuring data, but with apologies, am confused about the accompanying text.

Thus:

a) Supplementary Figure 3a shows CF cells. My expectation was that the miR-9 mimic would DECREASE luciferase-3' UTR ANO1 activity, and I believe this is what the figure shows. The accompanying text tells me that there was a significant INCREASE.

b) Supplementary Figure 3b shows non-CF cells. My expectation was that the miR-9 inhibitor would INCREASE ANO1 activity, and I believe this is what the figure shows. The accompanying text tells me there was a significant DECREASE.

Response: We thank the reviewer for the comment and have changed the mistaking text (lines 112-113 page 3-4).

Also, the natural place to refer to each component of the new data would be next to the data for the already inserted data. Thus, rather than a sentence added at the end of the paragraph covering both sets of new results, I would suggest the reader would benefit from keeping all mimic data together (CF and non-CF) and all inhibitor data together (CF and non-CF).

Response: We thank the reviewer for the comment, in the main figures of the article, the mimic and inhibitor data are merely grouped in the figure 3 in order to show the ANO1 direct regulation by miR-9 in CF and non-CF cells.

Response to Reviewer #2:

Reviewer #2 (Remarks to the Author):

The authors have addressed all of my comments. That said, I would encourage them to mention the proliferation effect observed between 4-8 hours as something that requires attention as this drug is developed further (perhaps around page 6 line 254.)

Response: We thank the reviewer for his comment, we have now mentioned the fact that proliferation is observed after 4h of wound healing experiment (page 6-7 lines 254-258).

Response to Reviewer #3:

Reviewer #3 (Remarks to the Author):

The response to the first point of criticism is not satisfying. It is important to know in which

cells ANO1 is upregulated by the TSB treatment. The ANO1 activity could be upregulated in the airway epithelial cells or for example in club cells. Increased ANO1 activity in epithelial cells could compensate CFTR deficiency, but upregulation of ANO1 in club cells could produce more mucus like in the asthma model shown before (Huang F et al, 2012). Therefore, it should be checked by immunostaining if TSB treatment increases ANO1 expression on the apical membrane of airway epithelial cells (see Huang et al).

Response: We would like to thank the reviewer for this critical comment. As previously described, the expression of ANO1 in the airway is mainly present in the ciliated cells (Huang et al. 2012) and in Goblet cells (Gorrieri et al. 2016) but modest in sub-mucosal glands (see below).

Figure 1: ANO1 immunostaining (green) in bronchial epithelial tissue

Mucus is an important point to investigate because the main cause of mortality for CF patients is due to chronic obstruction of the airways. For this point, we have combined two different approaches. First, we have “quantified” the main mucins expression in CF airways by immunostaining and by RT-qPCR. Second, we have quantified mucus clearance as suggested below in the last comment.

For the first point, by immunostaining quantification, we have not obtained a clear effect of the TSB on MUC5B in some specific cells (club cells, glandular cells...). In both models, in ALI and mice trachea, no clear overexpression was detected, but by this method, it is difficult to compare two conditions in two different slides except if a clear difference is present or/and different localization is observed (e.g., Huang F et al. 2012).

So, we have decided to perform RT-qPCR in the different isolated models present in the laboratory (CFBE41o-, KM4 cells, CUFI cells, primary human bronchial gland cells and mice trachea). In our condition, the secretory cells (KM4) have an increase of MUC5AC expression, but this increase is not significant (Fig. 2 below). For this reason, we have added this comment in data not shown in the text (page 5, lines 188-191).

Figure 2: Relative expression levels of MUC5AC mRNA in different cells treated with ANO1 TSB or control TSB. Data are quantified by qRT-PCR, normalized to GAPDH and presented as a fold-change compared to normalized controls. Data are presented as the mean \pm SD and were compared using Student's t-test.

For the second point, we have performed new experiments described below in the last comment.

The Iodide/YFP quenching assay measures the Iodide permeability induced by UTP, this is correlated to the increased activity of chloride channels but not to the chloride efflux, mentioned sometimes in the manuscript. Chloride efflux or influx is dependent on the membrane potential and on the intra- and extracellular chloride concentrations, which are not measured by this assay.

Response: We agree with the reviewer that this technique allows to measure the iodide permeability and not directly the chloride efflux. However, this technique is recognized and has been developed by the Verkman team and is widely used to study chloride efflux.

- [1] Bozoky Z, Ahmadi S, Milman T, Kim TH, Du K, Di Paola M, Pasyk S, Pekhletski R, Keller JP, Bear CE, Forman-Kay JD: Synergy of cAMP and calcium signaling pathways in CFTR regulation. Proc Natl Acad Sci U S A 2017.
- [2] Galiotta LJ, Haggie PM, Verkman AS: Green fluorescent protein-based halide indicators with improved chloride and iodide affinities. FEBS Lett 2001, 499:220-4.
- [3] Galiotta LV, Jayaraman S, Verkman AS: Cell-based assay for high-throughput quantitative screening of CFTR chloride transport agonists. Am J Physiol Cell Physiol 2001, 281:C1734-42.

The data of the YFP quenching assays are confusing: The axis of the graphs are differently labelled: Y axis with and without percentage, X axis with and without time (s). The summaries are not fitting to the original curves, for example in Fig. 6a the value of the control in the summary is 100 dF/min or 1,6 dF/sec, for TBS ANO1 dF300/min or 5 dF/sec. Compared to Fig 6d: WT mice 1,6 dF/sec and F508 TSM ANO1 mice 2 dF/s. The original curves in 6a show for control approximately changes of 1%/s and 15%/s for TSB ANO1 and 6D show for WT mice approximately 5%/sec and for CF TSB ANO1 more than 15 %/s. In Fig 4c the initial rate of Iodide uptake for 16HBE and CFBE TSB ANO1 cells are more or less the same 100 dF/min. The initial slope of the originals on the left side are different. For

16HBE cells approximately 5%/s and for TSB ANO1 15%/s. To avoid confusion: original values, not normalized values, should be shown for the original traces. In addition, the labels of the curves and of the summary are in some cases different.

Response: We thank the reviewer for his comment and have now corrected the graphs in order to homogenize the different graphs, we have so modified the axes of fig. 5d, fig. 6d et supplementary fig. 7.

The MQAE data shown are not conclusive. The change of intracellular chloride concentration should be shown before and after stimulation with cAMP or UTP.

Response: We agree with the reviewer that MQAE data alone are not conclusive; it was just a more classical method to confirm iodide uptake obtained with Halide sensor method. That is the reason that we have not included the result in this article. It was also to answer for a point of one reviewer. The protocol is described in the article published in 1998 by our group (Tabary et al., 1998).

To claim an alternative strategy to correct chloride efflux in CF patients, *in vivo* data are helpful. At least the effect of the TBS treatment could be easily confirmed by *ex vivo* experiments. Instead of CF cells, the trachea could be used in the same way as described in the manuscript to study the mucus dynamics. For this, the trachea could be mounted in a humidified chamber with basolateral perfusion. Movement of beads could be investigated before and after addition of apical UTP on the trachea from control and TBS treated mice.

Response: We thank the reviewer for his comment and have now added a new proof of *in vivo* effectiveness of TSB treatment. As suggested, we have studied the mucus dynamics in the trachea of CF mice *in situ* and *ex vivo* looking at the transport of phenol red dye in the trachea from the caudal part to the laryngeal part. Additionally, we studied the mucus dynamics using fluorescent beads on the trachea *ex vivo*. All of these experiments have shown a significant increase of the mucus dynamics in the trachea of CF mice treated with ANO1 TSB compared to CF mice treated with the control (Fig. 6e and 6f). We have added these results in the new version of the article (page 5, lines 191-202).

Taken together, all our data suggested that the increase of ANO1 expression, as same level as non-CF cells, is not sufficient to induce a significant over-expression of mucin, but is enough to increase mucus dynamics. Based on these results, we propose ANO1 TSB as an alternative strategy to correct chloride efflux in CF patients.

REVIEWERS' COMMENTS:

Reviewer #3 (Remarks to the Author):

The response to the points of criticism and the changes in the manuscript are now satisfying.

Response to reviewer #3

Reviewer #3 (Remarks to the Author):

The response to the points of criticism and the changes in the manuscript are now satisfying.

Response: We thank the reviewer for all the comments during the submission process. The different comments have increased the quality of this article, and we are happy that this manuscript is now satisfying for the reviewer.